# Mitochondrial Integrity Is Critical in Right Heart Failure Development

**DOI:** 10.3390/ijms241311108

**Published:** 2023-07-05

**Authors:** Marion Müller, Elfi Donhauser, Tibor Maske, Cornelius Bischof, Daniel Dumitrescu, Volker Rudolph, Anna Klinke

**Affiliations:** 1Agnes Wittenborg Institute for Translational Cardiovascular Research, Herz- und Diabeteszentrum NRW, University Hospital of the Ruhr-Universität Bochum, 32545 Bad Oeynhausen, Germany; mamueller@hdz-nrw.de (M.M.);; 2Clinic for General and Interventional Cardiology/Angiology, Herz- und Diabeteszentrum NRW, University Hospital of the Ruhr-Universität Bochum, 32545 Bad Oeynhausen, Germany

**Keywords:** right heart failure, mitochondria, oxidative stress, pulmonary hypertension

## Abstract

Molecular processes underlying right ventricular (RV) dysfunction (RVD) and right heart failure (RHF) need to be understood to develop tailored therapies for the abatement of mortality of a growing patient population. Today, the armament to combat RHF is poor, despite the advancing identification of pathomechanistic processes. Mitochondrial dysfunction implying diminished energy yield, the enhanced release of reactive oxygen species, and inefficient substrate metabolism emerges as a potentially significant cardiomyocyte subcellular protagonist in RHF development. Dependent on the course of the disease, mitochondrial biogenesis, substrate utilization, redox balance, and oxidative phosphorylation are affected. The objective of this review is to comprehensively analyze the current knowledge on mitochondrial dysregulation in preclinical and clinical RVD and RHF and to decipher the relationship between mitochondrial processes and the functional aspects of the right ventricle (RV).

## 1. Introduction

Right ventricular dysfunction (RVD) is an independent determinant of mortality in several cardiovascular disorders of right ventricular (RV) pressure or volume overload such as pulmonary arterial hypertension (PAH), congenital heart disease, left heart failure (LHF), or tricuspid valve regurgitation (TR) [1]. Whereas the current understanding of and treatment options for LHF have continuously improved during recent years, no therapeutic options are available in today’s clinical routine that specifically target the right heart [2,3,4]. This implies that significant interventricular differences exist, which are responsible for unsuccessful pharmacological treatment. Besides differences in morphology and hemodynamics, distinct cellular and molecular processes and their regulation under pathophysiological conditions may significantly account for individual maladaptive mechanisms of the right and left ventricles [3,5]. Whereas the principal pathogenic processes in right heart failure (RHF) are the same as in LHF, their importance and order may be diverse. Importantly, structural remodeling may play a minor role in RHF compared to LHF [6,7,8].

Given that the heart is the most metabolically active organ in the body, it is particularly reliant on mitochondrial integrity. Thus, mitochondrial dysfunction is suggested to play a prominent role in heart failure. Herein, we will review the current knowledge on mitochondrial damage, mitochondrial (metabolic) dysfunction, and the mitochondrial release of reactive oxygen species (ROS) and its interplay with the cardiomyocyte contractile function in RVD and RHF and discuss potential therapeutic options. Given the complexity of preclinical model phenotypes, mitochondrial functional characteristics, and oxidative stress, we will also pay attention to methodological issues.

## 2. The Powerhouse of the Heart

Mitochondria are oval organelles around 0.7 to 1 µm in size containing a double membrane structure with transversally oriented cristae formed by infolding of the inner membrane. They are tightly connected and form chain-like structures assembled in mitochondrial clusters. In the cardiomyocyte, mitochondria can be perinuclear, interfibrillar, and subsarcolemmal and possess different proteomic and physiological statuses [9,10]. Although mitochondria exhibit a dynamic characteristic, they are the source of energy generating the majority of the adenosine triphosphate (ATP) in the heart. The perpetual energy demand in the heart requires a mitochondrial volume of 23 to 32% of the total cardiomyocyte volume. The cardiac mitochondrial volume increases continuously from humans and dogs to rats, hamsters, and mice. Thus, the increased mitochondrial density correlates well with the increased heart rate and oxygen consumption in smaller animals [11].

In mitochondria, energy is generated via oxidative phosphorylation (OXPHOS). This complex process can be described as a sequential passage of electrons from negative (nicotinamide adenine dinucleotide, NADH, and flavin adenine dinucleotide, FADH_2_) to positive (molecular oxygen O_2_) redox potentials down the electron transport chain (ETC) in aerobic respiration. OXPHOS normalized to the number of mitochondria reflects the generation of energy in terms of ATP and is defined as a mitochondrial function. The ETC consists of four protein complexes located at the inner mitochondrial membrane. The main energy conversion reactions take place at complex I (NADH dehydrogenase), complex III (cytochrome c reductase), and complex IV (cytochrome c oxidase), which use the energy of the stepwise electron transfer for active pumping of hydrogen ions (H^+^) out of the mitochondrial matrix into the intermembrane space. Thereby, an electrochemical gradient is established across the inner mitochondrial membrane. The translocation of H^+^ back into the mitochondrial matrix through the F_1_F_O_ ATPase (ATP synthase complex) is coupled to the phosphorylation of ADP to generate ATP. The reduced coenzymes NADH and FADH_2_, the electron donors for the ETC, are produced in the tricarboxylic acid (TCA) cycle within the mitochondrial matrix. One enzyme of the TCA cycle, succinate dehydrogenase or complex II, is part of the ETC located in the inner mitochondrial membrane. Mitochondrial function can be determined by respirometry, which measures the O_2_ consumption rate in permeabilized muscle fibers, cardiac tissue samples, or isolated mitochondria after the addition of saturating concentrations of metabolic substrates and ADP. Of note, even by using living cells, the information is limited to the general function of the ETC protein complexes, their connectivity and coupling to the ATPase, and the lack of data on the availability of NADH, FADH_2_, ADP, or O_2_.

## 3. Redox-Optimized ROS Balance in the Heart

The ETC releases the superoxide radical (O_2_^•−^), a reactive oxygen species (ROS), even under physiological conditions, given that a small percentage of electrons prematurely reduce O_2_ at complex I and complex III. This production of mitochondrial ROS (mitoROS) is balanced by an antioxidant system located in the mitochondrial matrix [12]. The manganese-dependent superoxide dismutase (mitochondrial isoform: SOD2) converts O_2_^•−^ to O_2_ and hydrogen peroxide (H_2_O_2_), which is then further reduced to water by glutathione peroxidase (GPX) and peroxiredoxin (mitochondrial isoforms: PRX3 and PRX5). For the regeneration of GPX and PRX, the pyridine nucleotide NADPH is needed, whose reduction in mitochondria depends on TCA cycle activity. Therefore, TCA cycle enzyme integrity is essential for ATP production and the regeneration of antioxidative pathways. Pathological conditions can lead to dysfunction of the TCA cycle and OXPHOS in mitochondria, which effects ATP production and the generation of mitoROS due to an altered redox balance (NAD^+^/NADH, FAD/FADH_2,_ NADP^+^/NADPH ratios).

Under physiological conditions, the emission of mitoROS in the ETC is low, allowing the antioxidant system to balance ROS levels sufficiently with adequate NADPH availability for the regeneration of the antioxidant enzyme systems. However, under pathophysiological conditions, the ETC becomes uncoupled, resulting in a decreased membrane potential and less ATP generation. An extreme oxidized state, probably due to altered metabolic flows and less TCA cycle activity and therefore lower restoration of reduced coenzymes NADH and FADH_2_, depletes H_2_O_2_ scavenging capacity and the net ROS emission is enhanced despite eventually lower ROS production (Figure 1).

In addition to the generation of O_2_^•−^ or H_2_O_2_ in mitochondria, there are other intracellular sources of ROS production: monoamine oxidase (MAO), xanthine oxidase (XO), uncoupled nitric oxide synthase (NOS) and NADPH oxidase (NOX). MAOs are located in the outer mitochondrial membrane deputed to inactivate neurotransmitters, like serotonin, thereby producing H_2_O_2_ [13]. Endothelial NOS generates nitric oxide (NO) in the vascular endothelium to regulate vascular tone but is also expressed in cardiomyocytes. Uncoupled dysfunctional NOS produces O_2_^•−^, contributing to increased ROS [14]. The NOX isoforms are membrane-bound, ROS-producing enzymes. While most NOX isoforms produce O_2_^•−^, NOX2 and NOX4, both expressed in cardiomyocytes, generate H_2_O_2_. During the discovery of NOX, the physiological role of ROS was highlighted, taking into account that many redox-dependent biological processes are fine-tuned by ROS production and antioxidant enzyme systems: SOD, GPX, PRX, coupled NOS, and catalase (CAT) [15].

An unbalanced action of ROS, which is not buffered by the antioxidant system, can result in DNA, protein, and lipid modifications, a process which is defined as oxidative stress (Figure 1). DNA exposed to ROS can be modified to guanine, allowing it to pair with cytosine and adenine leading to double-strand breaks and genetic instability. In general, this can affect nuclear and mitochondrial DNA (mtDNA). However, due to the circular nature of mtDNA and cell-cycle-independent replication as well as the high copy number, mtDNA is more prone to oxidative damage. mtDNA fragmentation can be detected by quantitative real time PCR via assessment of the ratio of stable amplified long and shorter fragments of mtDNA [16]. Oxidative DNA modification, such as 8-hydroxy-2′-deoxyguanosin (8-OHdG), is an accepted marker of ROS and can be measured by various standard techniques such as immunohistochemistry or high-pressure liquid chromatography (HPLC). Oxidative post-translational modifications of proteins are often associated with functional inactivation or even damage of the affected proteins. These include protein carbonylation, detected by 1,3-dinitrophenylhydrazine (DNPH)-based *Oxyblots* [17], dityrosine, nitrotyrosine, and nitrocysteine formation, measured with specific antibodies or mass spectrometry, oxidation of thiol residues such as glutathionylation, and the formation of disulfides or sulfenic, sulfinic, and sulfonic acids, which can be detected by native gel electrophoresis and specific antibodies. For the quantitative detection of oxidative protein modifications in excised myocardial tissue, it needs to be considered that some oxidative protein modifications are reversible, like disulfid formation, glutathionylation, and sulfenic acid formation, and might be reduced during sample preparation processes. Likewise, ex vivo oxidative modifications can develop during tissue processing and will result in false positive results. Nitrotyrosine and dityrosine formation, carbonylation, and the formation of sulfinic and sulfonic acids from thiol groups of proteins are irreversible. In addition, there are several mass-spectrometry-based techniques available to specifically detect oxidative modification of proteins in comprehensive screening approaches [18,19]. Furthermore, the detection of lipid peroxidations, like 4-hydroxynonenal (4-HNE) or malondialdehyde (MDA), is a widely used method to indirectly detect increased ROS formation in tissues. Of note, all the explained methods detect modifications, which occur as a consequence of increased cellular ROS. Direct measurements of O_2_^•−^ and H_2_O_2_ are possible in tissue or isolated mitochondria by using, for example, fluorescence-dye-based techniques. However, it needs to be considered that mitoROS release in excised cardiac tissue does not absolutely reflect the in vivo situation. To conclude, ROS occur naturally in the myocardium and are important signal mediators. In contrast, oxidative stress impairs cellular function and is associated with pathological conditions.

## 4. Mitochondrial Integrity Is Impaired in Right Heart Failure

In order to assess the significance of mitochondrial function in the development of RHF, the experimental model system needs to be considered critically. Experimental studies measuring ROS or oxidative modifications in RV myocardium in different RVD/RHF models are summarized in Table 1 with a special focus on the postulated mechanism leading to ROS generation and functional consequences.

As RHF is a leading cause of mortality in PAH patients, many experimental studies have used rodent models of PAH, induced either by injection of the vascular endothelial growth factor (VEGF) inhibitor, Sugen 5416, followed by hypoxia (SuHx) or the injection of monocrotaline (MCT), a pneumotoxic pyrrolizidine alkaloid. Both model systems are characterized by pulmonary arterial (PA) remodeling and constriction with increased PA pressure and pulmonary vascular resistance leading to compensated RV hypertrophy and eventually RHF. In contrast to the PAH rodent model systems, the model of PA banding (PAB) is widely used to study RV remodeling independently of pulmonary vascular disease. Of note, different mouse wild-type strains might react differently to RV pressure overload, either developing compensated RV dysfunction or failure [6], which was also observed by Shults et al. [29] in SuHx Sprague Dawley and Fisher rats. It might be obvious that the degree of afterload triggered by the degree of PA stenosis in PAB rodent models is associated with the extent of RV dysfunction and structural remodeling [6,37]. However, to assess the molecular mechanisms mediating the transition of compensated RV hypertrophy (RVH) and RVD to RHF, it is essential to determine the signs of RHF in experimental rodent models. Signs of congestion reliably reflect the presence of RHF, as presented in experimental studies, i.e., liver weight [31,32,33,34], macroscopic signs of a nutmeg-colored liver [6], and ascites. In addition, impaired exercise capacity [34] and/or mortality [6,24,29] can reflect RHF. A quantitative parameter to determine signs of RHF might be the morphometric assessment of congestion-modified areas in haematoxylin/eosin-stained liver sections. The extent of hepatic congestion showed a strong positive correlation with right atrial (RA) area and a strong negative correlation with systolic RV function assessed by echocardiography in PAB mice [6].

To understand RHF, it is important to consider that already under physiological conditions, the RV differs from the left ventricle (LV) in loading conditions and morphology and displays fundamental differences on the cellular and molecular levels. There is accumulating evidence that the RV is more susceptible to oxidative stress probably because of a less adaptable antioxidant enzyme system. In healthy rats, Nagendran et al. [38] measured a lower mitochondrial membrane potential in RV cardiomyocytes and RV tissue compared to the LV, which might provide evidence for a lower activity of the ETC complexes in RV compared to LV tissue. Furthermore, the membrane potential rose with disease severity in a mouse model of PAH induced by MCT [38], indicating adaptation of RV cardiomyocytes to increasing workloads. Interestingly, Phillips et al. detected an identical expression pattern and relative amount of proteins, the same amount of post-translational protein modifications, and an identical amount of mitochondria in RV and LV tissue of healthy rabbits and pigs [39]. Given the lower oxygen consumption and lower ATP generation rate [40,41] of the resting RV, the RV should hold larger metabolic reserve capacities than the LV. This should imply lower mitoROS generation and a less vulnerable RV under an increasing workload. But in contrast to the LV showing enhanced antioxidative capacity under the progression of pathological conditions [22,23,36], the antioxidant enzyme system is unchanged or is even decreased in the RV myocardium of mice after RV pressure overload [6,34], rats with chronic NO deficiency [36], and in human end-stage HF patients secondary to ischemic HF or idiopathic dilated cardiomyopathy [21,22,23]. To detect the status of the antioxidative system in the RV, several preclinical studies have measured SOD expression on mRNA levels [6,34,36] with differentiation between *Sod1*, an isoform mainly expressed in the cytosol, and *Sod2*, the mitochondrial isoform. In studies using human RV myocardium, CAT activity was determined [22,23] in the tissue of end-stage LHF patients.

Progressive destruction of the mitochondrial network, which may result in impaired formation of ATP due to decreased OXPHOS under aerobic conditions, is associated with the development of RHF (Figure 1). A decreased amount of mitochondria has been shown in experimental PAH [25,32,42] and RV pressure overload [6,34] by detecting decreased activity of citrate synthetase (CS), the pace-maker enzyme of the TCA cycle in mitochondria, or decreased amounts of mtDNA. Experimental data could be confirmed in RV tissue samples of congenital heart disease patients divided into two groups of compensated RVH and RHF based on postoperative cardiac magnetic resonance imaging (MRI) [43]. While CS and succinate dehydrogenase activities were maintained in RVH, they were significantly decreased in RHF. The mtDNA copy number progressively decreased during the transition from RVH to RHF and inversely correlated with RV systolic pressure [43]. In contrast with reduced mitochondrial biogenesis, reflected by decreased *Pparagc1a* (PGC1α) mRNA levels, reported in SuHx rats [25], Karamanlidis et al. showed significantly increased *Pparagc1a* (PGC1α) mRNA expression in human RVH and again normalized mRNA levels in human RHF [43]. This might indicate mitochondrial biogenesis as a highly dynamic, compensatory effect during the progression of RVH/RVD to RHF. The mitochondrial architecture is also influenced by impaired mitochondrial fission and fusion (Figure 1), a process by which mitochondria divide or fuse to maintain their integrity and function [44]. Of note, increased mitochondrial fission as well as enhanced fusion are reported in experimental PAH [31,45,46,47]. The fission/fusion ratio and thereby mitochondrial biogenesis might also depend on disease severity, as Hwang et al. [34] showed significantly increased mitochondrial fusion in compensated RVH upon RV pressure overload by protein expression of mitochondrial dynamin-like GTPase (OPA1). Both OPA1 isoforms were significantly decreased in RHF upon RV pressure overload, while the mitochondrial fission protein, dynamin-related protein (DRP1), was significantly increased in isolated RV mitochondria in RHF upon PAB [34]. The same expression pattern, which ultimately leads to an impaired fission/fusion ratio in RVH, was also shown by Wüst et al. [42] in MCT rats. After injection of low-dose MCT resulting in compensated RVH, increased fusion and decreased fission were observed, while rats injected with a higher MCT dosage suffered from RHF and showed disrupted fission as well as fusion [42]. In accordance, data from transmission electron microscopy (TEM) of RV tissue showed an elevated mitochondrial area and increased mitochondrial size and higher amounts of mitochondrial clusters with a decreased number of mitochondria per cluster in the RV of SuHx rats at early time points. In contrast, at later time points, the mitochondrial area significantly decreased and mitochondrial integrity was disrupted, which was associated with increased ROS generation and reduced energy production in the RV tissue of SuHx rats [29]. An impaired mitochondrial ultrastructure was also shown in other PAH rodent models [25,32,42], models of RV pressure overload [25,34], and in human RV tissue of congenital heart disease patients [43].

## 5. Mitochondrial Oxidative Stress Mediates Right Heart Failure

Oxidative stress is suggested to contribute to the development of right ventricular dysfunction (RVD) and the transition from RVD to RHF [41,48,49]. In experimental models, significantly increased ROS generation in the RV myocardium has been frequently described [6,21,22,23,24,25,26,27,28,29,30,31,33,34,35,36,50], while in a minor number of studies, an association of increased ROS generation with RV function [6,23,24,25,26,30,31,33,34,35,36,50] or RHF [6,24,31,33,34] has been disclosed (Table 1). Zimmer et al. showed significantly increased H_2_O_2_ levels measured by oxidized phenol red absorbance in the RV myocardium of MCT rats already at early disease stages, while RV systolic function declined later [33]. This is in accordance with our own data in the PAB mouse model which presented significantly increased hyperoxidized PRX in RV already after one week. Interestingly, antioxidant treatment prevented RHF through a reduction in RV remodeling but did not change increased PA pressure or PA remodeling in SuHx rats [24] Accordingly, Redout et al. [31] showed that antioxidant therapy with the SOD/CAT-mimetic EUK124 attenuated RV hypertrophy and systolic dysfunction, but PA remodeling was not affected in MCT rats. In contrast, Bogaard et al. [24] observed significantly increased PA structural remodeling, ROS generation and RHF-dependent mortality in SuHx rats, but not in PAB rats. A causal relationship between mitoROS and RV systolic function is further supported by the data of Gomez-Arroyo et al. [25], showing significantly increased ROS generation in the RVs of SuHx rats going along with enhanced mRNA expression of *Pparagc1a* (PGC1α), the mitochondrial master regulator. Interestingly, the level of *Pparagc1a* (PGC1α) correlated with RV systolic function [25]. Of note, enhanced production of mitoROS was associated with mitochondrial dysfunction characterized by decreased oxygen (O_2_) consumption rates in the RV tissue of PAH rat models induced by SuHx [25] or MCT [32]. Hwang et al. [34] presented the involvement of mitochondrial dysfunction in the progression of RHF in a mouse model of PAB. They distinguished between RVD and RHF via exercise capacity in mice exposed to RV pressure overload and showed decreased transcription of ETC protein complexes, increased lipid peroxidation, and decreased oxygen consumption in the RV myocardium of mice with RHF compared to RVD [34]. Our group disclosed that mitoROS mediate the development of RHF following pressure overload in mice induced by PAB. In fact, C57BL/6J (6J) mice, which exhibit a genetic mutation leading to mitochondrial nicotinamide nucleotide transhydrogenase (NNT) deficiency [51] and thus generate fewer mitoROS under pressure overload, were protected from RHF compared to NNT-competent C57BL/6N (6N) mice [6]. The NNT is located in the inner mitochondrial membrane and regenerates NADPH to sustain antioxidative capacity [52,53] (Figure 1). It has already been disclosed for the pressure-overloaded LV [54] that the NNT acts in reversed mode under pathological conditions and requires the NADPH pool leading to increased mitoROS. In accordance with the LV data, 6N mice exhibited greater impairment in RV function, a more enlarged right atrial area, more pronounced venous liver congestion, and increased mortality upon RV pressure overload. Mechanistically, enhanced mitoROS and elevated apoptosis were detected in the RV myocardium of 6N mice compared to 6J mice upon PAB [6]. Interestingly, RV maladaptive remodeling reflected by RV fibrosis and hypertrophy were induced to the same extent in both mouse strains and were dissociated from the development of RHF upon RV pressure overload. This is in accordance with data from Hwang et al. [34], who used the same murine model of RV pressure overload and Boehm et al. [7], who analyzed a PAB mouse model with a self-dissolving suture [7]. In addition, anti-fibrotic therapy did not improve RV function in animal models of RV pressure overload and PAH [8]. In contrast, fibrosis correlates with systolic function in the LV and is an accepted predictor of mortality in left heart failure (LHF) [55,56,57,58]. The disconnection of fibrosis and hypertrophy with functional parameters in the development of RHF might reflect a pivotal difference of RHF and LHF pathomechanisms.

Subcellular mechanisms in RV cardiomyocytes might be more important than structural RV remodeling in the pathogenesis of RHF, which is also supported by earlier studies of Redout et al. [30]. The authors showed significantly increased NOX-dependent ROS generation as well as mitoROS generation in the RV tissue and isolated mitochondria of MCT rats to be associated with decreased active twitch force of the RV myocardium and the decreased oxygen consumption rate (OCR) of the RV mitochondria. The results describe the concept of ROS-induced ROS release (RIRR), meaning that O_2_^•−^, produced by NOX in early compensated RVD, might modify mitochondrial function with decreased ETC-dependent OXPHOS [30,32,34,42,59,60,61] resulting in increased generation of mitochondrial O_2_^•−^ [27,32,36]. RIRR resulting in enhanced levels of O_2_^•−^ in mitochondria is further amplified by the opening of ROS-sensitive channels such as the permeability transition pore (PTP) in the inner mitochondrial membrane, leading to depolarization of the mitochondrial network and impaired mitochondrial function. The vicious cycle again results in enhanced O_2_^•−^ production, which is then released and distributed via H_2_O_2_ throughout the myocardium [62] (Figure 1). The concept is confirmed by the data of Frazziano et al. [63], who presented enhanced *Nox4* mRNA expression in RV tissue after 1 h of PAB surgery leading to significantly increased H_2_O_2_ levels in mice after 6 h of PAB and, while the O_2_^•−^ levels were not affected [63]. Alzoubi et al. [26] also detected enhanced O_2_^•−^ production by using DHE fluorescence after in vivo injection in the RV myocardium of SuHx rats [26]. Translational studies further support NOX-dependent ROS release in the compensated disease stages of RVH/RVD, as shown by the data of Borchi et al. [22] and Nediani et al. [21] detecting increased O_2_^•−^ in RV tissue in end-stage LHF patients. In addition, Manni et al. [23] showed a correlation of PA pressure with RV protein carbonylation [23] in the same cohort of end-stage LHF patients, underlining an association between disease severity and oxidative damage in RV of patients with PH secondary to LHF. Our group found increased immunoreactivity for 8-OHdG, reflecting oxidative DNA modification in the RV tissue of explanted hearts in LHF patients with severely impaired RV systolic function and increased PA pressure compared to LHF patients with normal RV function and PA pressure [6].

An increase in mitoROS involved in the transition of RVD to RHF can be caused by mitochondrial dysfunction (Figure 1). The following paragraph will discuss therapeutic options to target mitoROS in RVD and RHF.

## 6. Therapeutic Options to Reduce Mitochondrial Oxidative Stress in Right Heart Failure

Pharmacological treatment options reducing ROS in RV myocardium are currently under investigation. Here, we focus on experimental studies directly targeting mitoROS (Table 2).

The antioxidant mitoTEMPO is a SOD mimetic and accumulates in mitochondria due to coupling to a cationic triphenylphosphonium molecule. It induces the dismutation of O_2_^•−^ to H_2_O_2_ [67]. Recent results from our group demonstrated that mitoTEMPO treatment in a preventive approach for up to 4 weeks significantly improved RV systolic function and reduced apoptosis in the RV myocardium of mice upon RV pressure overload. Importantly, reducing mitoROS prevented RHF in this mouse model of PAB, demonstrated by reduced hepatic venous congestion [6]. Although experimental studies have demonstrated improved LV function [68], restored mitochondrial respiratory capacity [68], and the prevention of ventricular arrythmias [69] after mitoTEMPO administration upon LV pressure overload, there are currently no clinical trials ongoing.

Another antioxidative component is mitoQ, which is a ubiquinone covalently attached to a cationic triphenylphosphonium molecule. It exerts its antioxidative potential, when ubiquinone is reduced to ubiquinol by complex II of the ETC, which then prevents oxidative damage like lipid peroxidation by O_2_^•−^ and the hydroperoxyl radical [70,71,72]. Preclinical studies focusing on the LV demonstrated that mitoQ reduced H_2_O_2_ production [73,74], preserved mitochondrial respiration [73], prevented cardiac fibrosis [74], and improved LV function [74] in LV pressure overload. In models of right heart diseases, mitoQ was found to attenuate acute hypoxic pulmonary vasoconstriction in PA and to inhibit PAB-induced RV dilation by decreasing the O_2_^•−^ concentration [64]. To quantify the O_2_^•−^ concentration in the heart homogenate, electron spin resonance spectroscopy was used. In addition, it was shown that RV dilation was significantly reduced, and RV systolic function was significantly increased after PAB in the mitoQ-treated mice [64]. MitoQ is currently being tested in clinical trials (for studies in cardiovascular diseases: NCT03586414, NCT05561556, NCT05410873, NCT03960073, and NCT02690064, see Table 3), but no results have been published yet.

The synthetic tetrapeptide SS-31 (Szeto-Schiller 31, elamipretide) acts selectively on mitochondria by restoring mitochondrial bioenergetics. It binds to cardiolipin, a phospholipid that is exclusively expressed in the inner mitochondrial membrane. Cardiolipin plays a central role in cristae membrane structure formation and organization of the respiratory chain components of the ETC [79]. Studies have shown that SS-31 protects against cardiolipin peroxidation, thereby preventing the loss of molecular interaction with cytochrome c [80,81]. This leads to the maintenance of mitochondrial cristae integrity and accelerates ATP recovery. It was demonstrated that ROS-dependent cell death was attenuated [80,82,83,84]. Currently, SS-31 is being tested in clinical trials for several diseases, including mitochondrial myopathies, aging, ischemia reperfusion injury, and heart failure (for studies in cardiovascular diseases: NCT02814097 and NCT02914665, see Table 3). In patients with stable heart failure with reduced ejection fraction (HFrEF), elamipretide was well tolerated but did not improve LV end systolic volume (ESV) after 4 weeks compared with the placebo [75]. The effects of SS-31 on the RV were investigated in a mouse model of PH induced by transverse aortic constriction (TAC). Administration of SS-31 for 60 days effectively attenuated the TAC-induced increase in RV systolic pressure and reduced BNP protein expression and mRNA expression of ROS-secreting proteins (NOX1/NOX2). However, given that in lung parenchyma of these animals, similar effects on the protein expression levels of NOX1/NOX2 and a reduction in oxidative damage were found, it is unclear whether SS-31 exerted a direct effect on the RV [65].

Small molecules such as the cysteine precursor N-acetylcysteine (NAC) or H_2_S donors are antioxidants that exert their effects by increasing glutathione (GSH) levels or stimulating the expression of thioredoxin. Thus, these molecules reinforce the antioxidative defense system [85,86,87,88,89]. A study by Chaumais et al. [90] revealed the beneficial effects of NAC treatment in a MCT-induced PAH model, showing significantly reduced PA remodeling, lung inflammation, RV hypertrophy, and fibrosis. In addition, NAC treatment significantly improved RV systolic function. However, ROS markers were not determined [90]. In accordance, Chen et al. [66] presented attenuated TAPSE and RV dilation going along with reduced RV hypertrophy and fibrosis in MCT rats [66]. Of note, Yazdi et al. [77] showed significantly improved RV systolic function in parallel with benefits in LV systolic function in HF patients with reduced ejection fraction (HFrEF) after 12 weeks of oral NAC treatment compared to the placebo [77]. The effects of H_2_S donors such as NaHS were analyzed in experimental models after myocardial infarction, showing improved mitochondrial integrity and decreased cardiac apoptosis [91]. The influences on RV myocardium have not been characterized, yet. Nevertheless, the clinical studies with the H_2_S donor SG1002 were considered safe in both healthy and HF patients (Table 3).

It is important to note that there are currently few studies addressing treatments that specifically target mitoROS in the RV. Further studies are needed to investigate the benefits of pharmacological treatment.

## 7. Mitochondrial and Contractile Function Are Connected in Right Heart Failure

Mitochondrial dysfunction including dysregulated energy generation may have a direct impact on the function and force generation of the cardiomyocyte. Significantly decreased OXPHOS was measured in permeabilized RV tissue [42,61] and RV fibers [32] of MCT rats and in isolated mitochondria of human RV tissue of PH patients [60]. All of these studies reported significantly lower aerobic complex I-coupled respiration in the presence of the energy substrates glutamate, malate, and pyruvate, as well as ADP [32,42,60,61]. Furthermore, Power et al. [32] tested O_2_ consumption in the presence of the energy substrates and ATP to measure the connectivity between myofibrillar ATPase and mitochondrial respiration. Endogenous turnover of ATP to ADP by cytosolic ATPases subsequently stimulates OXPHOS. ADP-limited OXPHOS was lower in the RV fibers of the MCT rats, going along with a significantly increased distance between the mitochondria and myofilaments detected by confocal microscopy in the RV tissue of the MCT rats [32]. This might indicate a mismatch between energy demand and supply in the hypertrophic RV cardiomyocytes of MCT rats with expanded myofilament content. In addition, mitochondrial creatine kinase (CK) expression was decreased in the RV of MCT rats [42,92], leading to reduced ATP delivery to the myofilament and a decreased ADP supply in the mitochondria (Figure 1). Gupte et al. [60] detected increased protein levels of ADP-ATP translocase (ANT3, ANT4) in the isolated RV mitochondria of PH patients compared to corresponding LV mitochondria [60], which might compensate for decreased mitochondrial CK to maintain ADP availability in the mitochondrial matrix. The impaired ADP/ATP exchange across the mitochondrial inner membrane contributes to alterations in OXPHOS capacity and mitochondrial membrane potential. Furthermore, altered TCA cycle activity favors a highly oxidized state with an increased NAD^+^/NADH ratio and FAD/FADH_2_ ratio resulting in mitoROS emission. This state is described as an “engine out of fuel” and is associated with impaired myocardial function, as confirmed by Wüst et al. [42] presenting force measurements and simultaneous detection of NADH and FAD autofluorescence in the RV trabeculae of MCT rats. Upon electrical stimulation at baseline levels, NADH levels were decreased in the RV trabeculae of the MCT rats and showed a correlation between the oxidized state (high levels of NAD^+^) and the severity of the PAH phenotype in the MCT model. The redox state of FAD showed the opposite course with high levels of reduced FADH_2_ in the RV of the MCT rats. During a rapid increase in electrical stimulation frequency, the NADH level declined and recovered much slower in the PAH trabeculae of the MCT rats compared to the controls, indicating uncoupling of contractile and mitochondrial function. In contrast, the FAD autofluorescence was strongly correlated with contractile output during changes in workload, indicating that mitochondrial dysfunction mainly contributes to complex I dysfunction [42]. Interestingly, adding pyruvate instead of glucose prevented NADH/FAD recovery upon the rapid increase in stimulation frequency [42], which might point towards less replenishment of metabolites from glycolysis and reduced TCA cycle activity. This is in accordance with increased hexokinase (HK) activity, catalyzing the initial reaction of glycolysis, in MCT rats [93] and ROS-dependent inhibition of aconitase, a TCA cycle enzyme [27]. The influences of metabolic substrate utilization on the principle functions of mitochondrial ETC, OXPHOS, and redox regulation as well as the interrelation with mitochondrial and contractile function are demonstrated in Figure 1.

To rule out that a lack of O_2_ provided to mitochondria contributes to the development of RHF, Balestra et al. [93] used an non-invasive technique, based on O_2_-dependent quenching of delayed fluorescence of mitochondrial protoporphyrin IX [94,95], to measure the mitochondrial O_2_ tension (oxygenation) in the RVs of MCT rats in vivo [93]. A higher number of mitochondria with an increased level of intramitochondrial O_2_ were detected in the RVs of MCT rats. In addition, increased HK activity and increased lactate dehydrogenase activity were observed in the early disease stages of MCT rats. Thus, decreased O_2_ consumption and metabolic adaptation, independent of hypoxia, might contribute to the progression of RHF [93]. In contrast, Wong et al. [59] showed inefficient O_2_ utilization, leading to RV ischemia/hypoxia in idiopathic PAH patients classified as New York Heart Association (NYHA) class III compared to IPAH patients of NYHA class II. O_2_ extraction fraction (OEF) and glucose uptake were measured non-invasively by a positron emission tomography-computed tomography scan (PET-CT) with corresponding tracers. RV function and hemodynamics were assessed by catheterization and cardiac MRI. Interestingly, the OEF of the RV tissue of the IPAH patients was nearly the same as the OEF of the LV tissue of healthy individuals, showing a much higher O_2_ extraction reserve in the RV compared to LV tissue at baseline. The IPAH NYHA class III patients showed significantly higher myocardial O_2_ consumption, no change in glucose uptake rate, and lower RV mechanical efficiency, which correlated with RV systolic function [59]. Mechanistically, the impaired mechanical efficiency might be partly explained by reduced atrioventricular coupling. Nevertheless, other factors sought in cardiomyocytes, which have been comprehensively described for LHF need to be considered to explain impaired contractile performance in the progression of RHF. In particular, oxidative damage of sarcomeric proteins has been reported [96,97,98]. Furthermore, altered calcium signalling and cellular electrophysiological changes, given that ATP-consuming ion channels such as the sarcoendoplasmic reticulum calcium ATPase (SERCA) and Na^+^/K^+^ pumps are dependent on mitochondrial integrity, might be of significant importance [99,100,101].

## 8. Substrate Metabolism Is Affected in Right Heart Failure

Mitochondria in the dysfunctional or failing myocardium lose their flexibility in substrate metabolism. Fatty acid oxidation (FAO) serves about 70% of energy generation in the adult healthy heart. Whereas energy yield per molecule is higher for fatty acids (FA) than for glucose via glucose oxidation (GO), oxygen consumption on the other hand is higher per molecule FA. Thus, the shift to GO is oxygen sparing, and even less oxygen is necessary for anaerobic glycolysis, which has the lowest energy yield compared to GO and FAO, however. Thus, under optimal conditions the cardiomyocyte can adapt substrate utilization in consideration of oxygen availability and energy demand. Driven not only by reduced perfusion and thus reduced oxygen supply but also by inflammatory and oxidative damage in heart failure as discussed above, FAO, TCA cycle activity, and OXPHOS can be decreased and glycolysis is enhanced, which results in reduced ATP generation [102]. These modifications resemble fetal conditions, and this fetal reprogramming is a hallmark of the compensatory process of the cardiomyocyte facing mechanical, hypoxic, metabolic, or other stress triggers and is usually associated with hypertrophy [103]. The shift towards anaerobic glycolysis despite the presence of O_2_ to generate ATP is called the *Warburg effect*, which is in part mediated by increased activity and expression of pyruvate dehydrogenase kinase (PDK), which inhibits pyruvate dehydrogenase (PDH) (Figure 1). Not only a potential lack of energy, but also the accumulation of toxic metabolites, in particular toxic lipid species, is a consequence of the metabolic shift. RHF is associated with, and very likely also causally related to, mitochondrial metabolic dysregulation, which becomes obvious in patient studies and preclinical models. In mouse or rat PH models, models of RV pressure overload induced by PAB, and a pig model of RV volume overload, decreased expression of genes and proteins involved in FAO was found in RV tissue [25,50,104,105]. Moreover, FAO itself was detected to be reduced [106], and accumulation of triglycerides or lipotoxic molecules such as ceramide was enhanced in RV tissue [104,105]. At the same time, GO was reduced in these models in RV tissue, as evident from attenuated PDH activity and enhanced PDK expression [107,108,109] and a reduction in OXPHOS [108,110,111]. Interestingly, the reduction in OXPHOS and impaired glucose responsiveness was also observed in the myoblast cell line H9C2 with a mutation in the bone morphogenetic protein receptor type 2 (BMPR2) [112], which is related to genetic PAH. Consequently, inhibition of PDH by dichloroacetate (DCA) improved RV oxygen consumption and cardiac output (CO) in MCT and PAB rat models [110] and enhanced RV systolic function and exercise capacity in the Fawn-Hooded rat model of PAH [107]. In pigs with RV volume overload, DCA induced an increase in OXPHOS in the RV and an improvement in RV ejection fraction under dobutamine stress [108]. In line with this, DCA infusion in the Langendorff-perfused hearts of rats with MCT-induced PAH increased inotropy in the RV of PAH but not control hearts and at the same time reduced lactate production in PAH hearts [38]. These findings suggest that enhancement of GO is protective in RV dysfunction. Given that, according to the Randle’s cycle, GO is enhanced when FAO is decreased, inhibition of FAO has also been tested as a therapeutic strategy. In a PAB rat model of compensated RV dysfunction, the FAO inhibitors trimetazidine and ranolazine both increased RV respiration and ATP generation and, at the same time, the cardiac index and exercise capacity of the animals [111]. However, at least for the LV, it has been questioned whether depression of FAO is useful or rather harmful in heart failure [103]. Instead, improvement in FA metabolism was shown to have protective effects on the RV. One potential therapeutic strategy is activation of the peroxisome proliferator-activated receptor gamma (PPARγ), which is a key modulator of glucose and lipid metabolism, including mitochondrial FAO [113,114]. It can be activated by thiazolidinediones like pioglitazone, which inhibited exaggerated RV glucose uptake in a SuHx rat model of PAH, reduced RV intramyocardial lipid accumulation, and normalized RV function [115]. It should be noted, however, that in end-stage RHF with low cardiac output and high end-diastolic ventricular filling pressures, this therapy might not be initiated, but oxygen sparing therapeutic approaches at the expense of less efficient ATP production may be considered instead of activating FAO. Another study used L-carnitine to improve FA metabolism in mice exposed to a high-fat diet and RV pressure overload by PAB or a BMPR2P mutation, which decreased RV lipid accumulation and improved RV function and hemodynamics [116]. Improvement of substrate metabolism can be further achieved through modulation of the activity of the 5′-adenosine monophosphate-activated protein kinase (AMPK), which, amongst other mechanisms, inhibits the acetyl-CoA carboxylase (ACC) and thereby increases FAO. The AMPK activator metformin reduced RV hypertrophy and improved RV diastolic function in a model of PAB combined with a Western diet [117]. In a rat MCT model, inhibition of the With No Lysine Kinase 1 (WNK1), which enhances AMPK activity, decreased RV lipid accumulation, increased FAO, and improved RV function and RV-PA coupling, independent of PA structural remodeling [104].

The identification of metabolic dysfunction in the RV of patients with PAH supports the concept of a causal role for RV dysfunction. Increased RV glucose uptake or RV to LV glucose uptake ratio was not only associated with high PA pressure, pulmonary vascular resistance, impaired RV systolic function, RV PA coupling, and exercise capacity [118,119,120,121,122,123] but was also an independent predictor for adverse prognosis [120,122]. The increase in glucose uptake goes along with altered FAO. Increased accumulation of lipids, long-chain free FA and ceramides in the RV, and long-chain FA in plasma were found in PAH patients [106,116,124]. In RV tissue from PAH patients with RHF, the expression of genes regulating FAO was decreased, which was accompanied by increased lipid deposition [105]. Whether targeting substrate metabolism in RHF can be successfully translated to patients remains elusive. In a phase II clinical trial treating 20 PAH patients with metformin for 8 weeks, RV fractional area change was significantly improved, whereas RV triglyceride content was decreased [125]. Further trials are warranted to advance the understanding of the pathophysiological role of mitochondrial metabolic function.

## 9. Conclusions

The presence of RVD and the transition from RVD to RHF is associated with mitochondrial dysfunction and enhanced mitoROS release. While mitochondrial respiration and ATP production can decrease with the progression of the disease, the demand for oxygen by the hypertrophied RV and the preference for glycolysis increase, reducing glucose and FA oxidation. This compromised cardiac energetics impairs RV contractility, which is further promoted by a reduction in CK expression and thus energy supply at the myofilament. Although these subcellular alterations are also observed in the development of LHF, the therapeutic relevance for preventing RHF might be more pivotal. In contrast to LHF, structural remodeling might play a minor role in RHF pathogenesis, which becomes obvious by the RV’s unique plasticity. Thus, the development of RHF largely involves rapidly reversible mechanisms with mitochondrial dysfunction, which compromises contractility, with this being among the critical processes. Despite equal energetic machinery available in the RV and LV, there are fundamental differences in the responses to increased workload. Under pathophysiological conditions, the RV seems to be more susceptible to oxidative damage than the LV due to the lower antioxidant capacity. Consequently, protecting mitochondrial function in combination with a decrease in afterload might emerge as a promising therapeutic concept. Some experimental studies directly targeting mitoROS in RV pressure overload and PAH have shown positive results, calling for further clinical studies.

## 10. Future Directions

It is unequivocal that the integrity of mitochondria is affected in the dysfunctional RV. Yet, the course and characteristics of mitochondrial dysfunction in RHF, and its causal relation and significance are elusive. Both a highly precise definition and description of the type of RVD and RHF in preclinical models, as well as a greatly elaborated and subtle methodological repertoire to assess mitochondrial functions and oxidative stress, are inevitable to advance our understanding of mitochondrial processes in the RV. For a translational perspective, we need to elucidate which mitochondrial mechanisms are responsible for which functional alterations of the RV, if any, and whether these have an impact on the clinical phenotype. Therapeutic targets need to be evaluated carefully and comprehensively with particular respect to the time course of the disease. In this context, it will be interesting to investigate whether mitochondria-targeted therapies might be effective not only to prevent the transition from RVD to RHF but also to support the decompensation of the RV upon elimination of the underlying cause of the disease, such as lung transplantation or repair of the tricuspid valve.

## Figures and Tables

**Figure 1 ijms-24-11108-f001:**
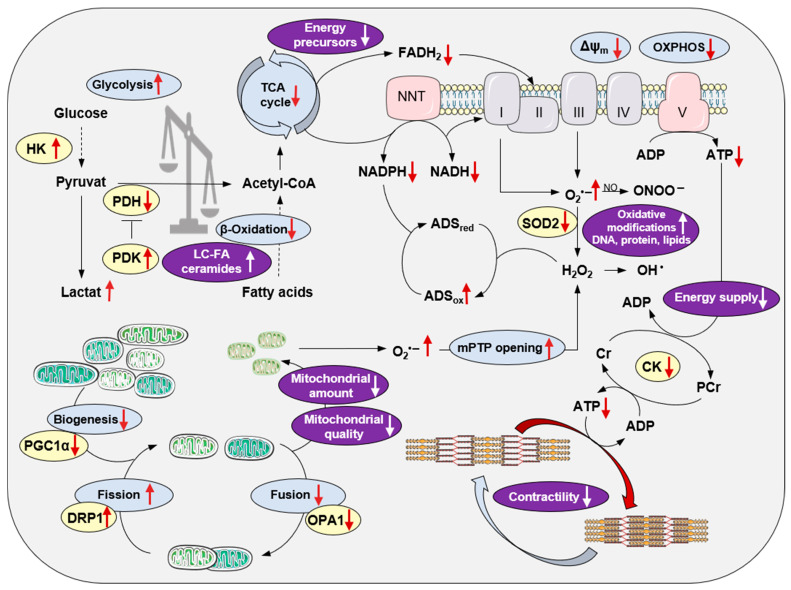
Modifications of cardiac mitochondrial integrity in right heart failure. Principal processes of energy metabolism, the mitochondrial electron transport chain, oxidative phosphorylation, redox regulations, mitochondrial biogenesis and dynamics, and their impact on the sarcomere are depicted. Red arrows indicate alterations in right heart failure. The purple color indicates the pathological consequences of mitochondrial disintegration. For better clarity, the reaction equations are simplified. **Abbreviations**: ADS, antioxidative defense system; Cr, creatine; CK, creatine kinase; DRP1, dynamin-related protein1; HK, hexokinase; LC-FA, long-chain fatty acids; mPTP, mitochondrial permeability transition pore; NNT, nicotinamide nucleotide transhydrogenase; OPA1, optic atrophy 1; OXPHOS, oxidative phosphorylation; PDH, pyruvate dehydrogenase; PDK, pyruvate dehydrogenase kinase; PGC1α, peroxisome proliferator-activated receptor γ coactivator 1α; SOD, superoxide dismutase; TCA, tricarboxylic acid.

**Table 1 ijms-24-11108-t001:** Experimental studies addressing oxidative stress and/or mitochondrial dysfunction in right heart dysfunction and failure.

Model/ Species	Main Finding ROS (Method)	Main Finding Mitochondrial Dysfunction	Postulated Molecular Mechanism	Association with PV Function? (Yes/No)	Association with RV Function? (Yes/No)	Refs.
**TOF** RVOT tissue Human	Severity of cyanosis (O_2_ saturation and hematocrit)	-	Chronic hypoxia in TOF associated with higher RV structural remodeling	no	no	[20]
**LHF** RV/LV tissue NF, DCM, ICM human	Increased ROS in RV compared to LV (NOX O_2_^•−^ lucigenin and MDA ELISA)	-	Less activation of antioxidative enzyme system in RV NOX-dependent O_2_^•−^ production is the main ROS source in the failing heart	no	no	[21,22]
**LHF** RV/LV tissue NF, DCM, ICM human	Increased ROS in RV not in LV (carbonylation Oxyblot and MDA ELISA)	-	MAO-dependent H_2_O_2_ source of ROS; the RV protein oxidation index correlates with pulmonary artery pressure	yes	no	[23]
**LHF** RV tissue DCM human	Increased ROS in RV (8OHdG IHC)	-	Increased mitoROS associated with impaired RV function	yes	yes	[6]
**PAH** PAB SuHx rat (male)	Increased ROS in SuHx compared to PAB RV (MDA IHC)	-	Antioxidant Nrf2/HO1 signaling prevents RV remodeling and RHF not PV remodeling	yes	yes	[24]
**PAH** SuHx rat (male)	Increased ROS in RV (8OHdG IHC)	Impaired mito morphology (TEM) Decreased mito copy number Decreased mito biogenesis Decreased OCR	Increased ROS and decreased mito biogenesis results in decreased OXPHOS in RHF	yes	yes	[25]
**PAH** SuHxNx rat (male)	Increased ROS in RV (DHE IF in vivo; nitrotyrosine IHC; NOX O_2_^•−^ lucigenin)	-	Reduction in NOX-dependent ROS production prevents RV structural remodeling and dysfunction	yes	yes	[26]
**PAH** SuHxNx rat (male)	Increased ROS in RV (NOX O_2_^•−^ lucigenin)	-	ROS-dependent inhibition of aconitase and altered pyridine nucleotide metabolism in mito	no	no	[27]
**PAH** SuOVA rat (male)	Increased ROS in RV (protein level: GSSG, nitrotyrosine, and nitrocysteine) Unchanged (protein level: MDA, 4-HNE, and carbonylation Oxyblot)	-	Increased oxidative modification in RV originates from xanthine oxidase	yes	no	[28]
**PAH** SuHxNx rat (male)	Increased ROS in RV (GSSG MS)	Impaired mito morphology (TEM) Decreased energy efficiency of mito (TEM)	Disruption of mitochondrial structure and reduced energy production associated with mortality	no	no	[29]
**PAH** MCT rat (female)	Increased ROS in RV (NOX O_2_^•−^ lucigenin and nitrotyrosine IHC)	Increased complex II protein level; reduced ubiquinone Decreased OCR	ROS-induced ROS release through initial NOX-dependent ROS generation associated with mito dysfunction	no	yes	[30]
**PAH** MCT rat (male)	Increased ROS in RV (nitrotyrosine IHC and carbonylation Oxyblot)	Increased mito fusion	Antioxidative treatment improves RV hypertrophy and function, but PV remodeling is not affected	yes	yes	[31]
**PAH** MCT rat (male)	Unchanged ROS (H_2_O_2_ AmplexRed)	Unchanged mito copy number Increased distance between mito- and myofibril Decreased OCR	Lower OXPHOS and impaired ATP supply to myofibrils	no	no	[32]
**PAH** MCT rat (male)	Increased ROS in RV not LV (H_2_O_2_ phenol red, TBARS content; sulfhydryl content)	-	Insufficient antioxidative system do not prevent early ROS production leading to RV structural remodeling and dysfunction	no	yes	[33]
**RHF** PAB FVB mouse (male)	Increased ROS in RV (4HNE protein level)	Impaired mito morphology (TEM) decreased mito copy number Impaired mito fission/fusion Decreased OCR	Decreased transcription of ETC complexes and increased ROS leading to mito dysfunction in RHF	no	yes	[34]
**RHF** PAB C57BL/6N mouse (male)	Increased ROS in RV (PRX-SO_2/3_ protein level and 8OHdG IHC)	Decreased mito copy number	Transition of RV dysfunction to failure dissociated from RV structural remodeling and hypertrophy; mitoROS trigger for RHF	no	yes	[6]
**PH** Cav^−/−^/Hx mouse	Increased ROS in RV (NOX O_2_^•−^ lucigenin)	-	RV structural remodeling and RHF independent of PA pressure	yes	yes	[35]
**Chronic NO deficiency** L-NAME, rat eNOS^−/−^ mouse (female)	Increased ROS in RV not LV (DHE IF, oxiTm, and peroxynitrate ELISA) Increased mitoROS in RV (MitoSOX IHC)	Unchanged mito biogenesis	Prevented antioxidative defense system in RV and increased ROS associated with increased RV structural remodeling and dysfunction	no	yes	[36]

**Abbreviation:** TOF, tetralogy of Fallot; RVOF, right ventricular outflow tract; RV, right ventricle; LV, left ventricle; NF, non-failing; DCM, dilated cardiomyopathy; ICM, idiopathic cardiomyopathy; LHF, left heart failure; PAH, pulmonary artery hypertension; OVA, ovalbumin; PAB, pulmonary artery banding; Su, sugen5416; Hx, hypoxia; Nx, normoxia; MCT, monocrotaline; FVB, friend leukemia virus B mouse strain; RHF, right heart failure; C57BL/6N, inbred mouse strain; PH, pulmonary hypertension; NO, nitric oxide; L-NAME, N^ω^-nitro-l-arginine methyl ester; eNOS, endothelial nitric oxide synthase; O_2_, oxygen; O_2_^•−^, superoxide; NOX, NADPH oxidase; MDA, malondialdehyde; ELISA, enzyme-linked immunosorbent assay; 8-OHdG, 8-hydroxy-2′-deoxyguanosin; IHC, immunohistochemistry; TEM, transmission electron microscopy; OCR, oxygen consumption rate; DHE, dihydroethidium; IF, immunofluorescence; GSSG, oxidized glutathione; 4HNE, 4-hydroxynonenal; ROS, reactive oxygen species; MS, mass spectrometry; H_2_O_2_, hydrogen peroxide; OXPHOS, oxidative phosphorylation; ETC, electron transport chain; PRX-SO_2/3_, peroxiredoxin sulfenic and sulfonic acid at cysteine; TBARS, thiobarbituric acid reactive substances.

**Table 2 ijms-24-11108-t002:** Experimental studies reducing mitochondrial oxidative stress in right heart dysfunction and failure.

Component	Experimental Model	Molecular Marker Improved/Method	Functional Marker Improved/Method	Main Results	Ref.
**mitoTEMPO**	PAB C57BL/6J C57BL/6N mice	Apoptosis/TUNEL	TAPSE/echocardiography Hepatic venous congestion/stain liver sections	MitoTEMPO treatment protects ROS-dependent RHF upon PAB.	[6]
**mitoQ**	Hx PAB C57BL/6J mice	Hx and PAB: O_2_^•−^ concentration/electron spin resonance spectrometry	Hx: RVWT; RVID; RVOTD/echocardiography Fulton/morphometric PAB: RVID; TAPSE/echocardiography Fulton/morphometric	MitoQ treatment reduces RV remodeling in chronic hypoxia and upon PAB. RV systolic function improved only upon PAB.	[64]
**SS-31**	TAC C57BL/6 mice	NOX1/NOX2 expression/protein level (lung) Protein carbonylation/Oxyblot (lung)	RV fibrosis/stain RV sections RVSP/RV catheter Vessel density and arterial muscularization/stain lung parenchyma Alveolar sacs/IHC	SS-31 treatment attenuates TAC-induced PH.	[65]
**Melatonin** **or** **NAC**	PAH MCT Sprague-Dawley rat	MDA fluorescence (RV)	RV hypertrophy/stain RV sections RV fibrosis/stain RV sections TAPSE, RVEDD, RV Area/echocardiography RVSP, TPR/PV loop	Melatonin or NAC reduced RV remodeling and improved RV function in PAH rats	[66]

**Abbreviation:** PAB, pulmonary artery banding; C57BL/6N, inbred mouse strain; TUNEL, TdT-mediated dUTP-biotin nick end labeling; TAPSE, tricuspid annular plane systolic excursion; ROS, reactive oxygen species; Hx, hypoxia; O_2_^•−^, superoxide; TAC, transverse aortic constriction; NOX1 and NOX2, NADPH oxidase; RVWT, right ventricular wall thickness; RVID, right ventricular internal diameter; RVOTD, right ventricular outflow tract diameter; RVSP, right ventricular systolic pressure; PH, pulmonary hypertension; IHC, immunohistochemistry; MDA, malondialdehyde; RVEDD, right ventricular end diastolic diameter; TPR, total pulmonary resistance; PV, loop pressure volume loop.

**Table 3 ijms-24-11108-t003:** Clinical studies addressing cardiovascular diseases.

Component	Treatment	Disease	Patient Cohort	Outcome/Status	Number Clinical Trail
**Elamipretide**	40 mg once daily for 28 ds	HF	Stable HF (NYHA II-III) LVEF < 45%	Elamipretide was well tolerated but did not improve LVESV at 4 wks in patients with stable HFrEF compared with the placebo	NCT02814097 [75]
**Elamipretide**	20 mg once daily for 7 ds	HF	Patients hospitalized with congestion due to HF LVEF < 40% NT-proBNP >1500 pg/mL	Unknown	NCT02914665
**mitoQ**	20 mg twice daily for 4 wks	Diastolic dysfunction due to ageing	Healthy patients between 50 to 75 years of age	Recruiting	NCT03586414
**mitoQ**	20 mg once daily for 8 wks	Hypertension	BP > 150/90 mmHg and BMI ≤ 40 kg/m^2^	Recruiting	NCT05561556
**mitoQ**	40 mg daily for 12 mths	DCM	LVEF < 45% plasma NT-pro-BNP > 250 ng/L	Recruiting	NCT05410873
**mitoQ**	20 mg daily for 4 wks	HFpEF	Stable HF (NYHA II-III) LVEF > 50%	Unknown	NCT03960073
**mitoQ**	10 mg on 2 ds separated by at least 72 h	CF	FEV1 percent predicted > 30% Resting O_2_ consumption > 90%	Active, not recruiting	NCT02690064
**NAC**	High dose twice daily for 2 ds	MI	ST-elevation infarction < 12 h Angina	No benefit with respect to CIN and myocardial perfusion injury	NCT00463749 [76]
**NAC**	600 mg twice daily for 12 wks	HF	Stable HF (NYHA II-III) LVEF < 45%	NAC significantly improved RV and LV systolic function in HFrEF compared to the placebo	IRCT20120215009014N333 [77]
**SG1002**	200 mg for 7 ds, then 400 mg for 7 ds, and then 800 mg for 7 ds	HF	Stable HF (NYHA II-III) LVEF < 40%	SG1002 was well tolerated, and patients showed increased/stable H_2_S level	NCT01989208 [78]

**Abbreviation:** HF, heart failure; NYHA, New York Heart Association classification; LVESV, left ventricular end systolic volume; HFrEF, heart failure with reduced ejection fraction; BP, blood pressure; BMI, body mass index; DCM, dilated cardiomyopathy; NT-proBNP, B-type natriuretic peptide; LVEF, left ventricular ejection fraction; CF, cystic fibrosis; FEV1, forced expiratory pressure in 1 s; MI, myocardial infarction; CIN, contrast induced nephropathy; H_2_S, hydrogen sulfide; O_2,_ oxygen; mths, months; wks, weeks; ds, days; h, hours; mg, milligrams.

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
