# Peer review of "Mitochondrial Integrity Is Critical in Right Heart Failure Development"

_ijms, 2023, doi:10.3390/ijms241311108_

Round 1

Reviewer 1 Report

In this review, the authors analyzed the relationship between mitochondrial dysregulation and right heart failure. Overall, it is well written. Here are my suggestions to improve it.

It is a pity that this review constrained their summary to the molecular level, ignoring HF-caused cellular electrophysiological changes. In HF, ion channels and action potential properties change (https://doi.org/10.1016/j.yjmcc.2012.11.020 & https://doi.org/10.1016/S0008-6363(02)00734-4). The link between mitochondrial dysfunction and HF-caused electrophysiological and pump activity changes can’t be ignored. In cardiomyocytes, there are ATP-sensitive K current and ATP-consuming pumps. After mitochondrial dysfunction, ATP level will reduce, leading to increased ATP-sensitive K current. The sarcoendoplasmic reticulum calcium ATPase (SERCA) pump acts to transport calcium ions from the cytosol back to the sarcoplasmic reticulum. Na/K pumps are responsible for clearing sodium influxes after action potentials. When ATP level is lowered, impaired SERCA causes arrhythmogenic APD alternans (https://doi.org/10.1155/2019/8237071), and impaired Na/K pump cause sodium accumulation (https://doi.org/10.1073/pnas.2105795118). These phenomena have been observed in experiments and theoretical studies (see above references).

The authors should expand the review to include the factors above.

Author Response

Response to reviewers’ comments

Reviewer 1

It is a pity that this review constrained their summary to the molecular level, ignoring HF-caused cellular electrophysiological changes. In HF, ion channels and action potential properties change. The link between mitochondrial dysfunction and HF-caused electrophysiological and pump activity changes can’t be ignored. In cardiomyocytes, there are ATP-sensitive K current and ATP-consuming pumps. After mitochondrial dysfunction, ATP level will reduce, leading to increased ATP-sensitive K current. The sarcoendoplasmic reticulum calcium ATPase (SERCA) pump acts to transport calcium ions from the cytosol back to the sarcoplasmic reticulum. Na/K pumps are responsible for clearing sodium influxes after action potentials. When ATP level is lowered, impaired SERCA causes arrhythmogenic APD alternans, and impaired Na/K pump cause sodium accumulation. These phenomena have been observed in experiments and theoretical studies. The authors should expand the review to include the factors above.

We thank the reviewer for her/his comment and fully agree, that alterations in cellular electrophysiology are of high relevance in the pathophysiological progression of all different forms of heart failure (HF). Our review article is focused on right ventricular dysfunction and molecular processes leading to failure of the right ventricle. We selected the most relevant mitochondria-related pathophysiological processes in terms of a) their uniqueness b) their difference and/or c) their relevance in right heart failure (RHF) development compared to other forms of HF. While considering these criteria we choose mitochondrial biogenesis, redox balance, substrate utilization and oxidative phosphorylation to be discussed comprehensively. We now mention the aspect of cellular electrophysiological changes in ll. 357 to be of importance in RHF: “Nevertheless, other factors sought in cardiomyocytes, which have been comprehensively described for LHF need to be considered to explain impaired contractile performance in the progression of RHF. In particular oxidative damage of sarcomeric proteins has been reported[97-99]. Furthermore, altered calcium signalling and cellular electrophysiological changes, given that ATP-consuming ion channels such as the sarcoendoplasmic reticulum calcium ATPase (SERCA) and Na/K pumps are dependent on mitochondrial integrity, might be of significant importance[100-102].”

Reviewer 2 Report

The review is exhaustive, timely and very informative.

1. Some sentences may be better rephrased for more clarity -1. Line 29

2. Line 172- Lung weight increase is a sign of left heart failure

3. Any role for ARNI ?

Author Response

Response to reviewers’ comments

Reviewer 2

The review is exhaustive, timely and very informative.

  1. Some sentences may be better rephrased for more clarity -1. Line 29
  2. Line 172- Lung weight increase is a sign of left heart failure

We thank the reviewer for her/his positive perception of our review. We rephrased the sentences accordingly.

  1. Any role for ARNI ?

We have discussed recently, which of the established HF drugs have an effect in RHF (http://dx.doi.org/10.21037/cdt-20-592). ARNI did not show any benefit on right heart function in rodent PAH or RV pressure overload models (Am J Respir Crit Care Med 2012;186:780-9; Circulation 2001;104:939-44). In patients with pulmonary hypertension, ACE inhibitors did not induce any benefits. These data point towards the fact that angiotensin signalling is not a suitable therapeutic target in RHF.

Reviewer 3 Report

The review titled "Mitochondrial Integrity is critical in right heart failure development" delves into the molecular processes underlying right ventricular dysfunction (RVD) and right heart failure (RHF). Understanding these processes is crucial for developing tailored therapies that can mitigate the growing mortality rate among affected patients. This review highlights the emerging significance of mitochondrial dysfunction as a pivotal factor in RHF development.The objective of this comprehensive analysis is to summarize the current knowledge on mitochondrial dysregulation in both preclinical and clinical RVD and RHF.

The review is an outstanding scholarly work, characterized by meticulous attention to detail and well-documented analysis. It explores the topic comprehensively. The engaging writing style guides the reader through complex concepts. Well-written and documented!

In the introduction, it would be great if the authors could include a figure that visually illustrates the structure of mitochondria and highlights its multiple roles. This visual representation would effectively enhance the reader's understanding of the complex and diverse functions performed by mitochondria within cells. By providing a visual aid, the authors can succinctly convey the significance of mitochondria in various cellular processes, setting the stage for a comprehensive exploration of their role in the context of the review's focus.

In addition, it would greatly enhance the manuscript to include illustrative figures for key sections such as "Redox optimized ROS balance in the heart," "Mitochondrial integrity is impaired in right heart failure," and "Mitochondrial oxidative stress mediates right heart failure", “Therapeutic options to reduce mitochondrial oxidative stress in right heart failure”. The inclusion of visual aids would effectively complement the written analysis and facilitate comprehension of these critical aspects.

Author Response

Response to reviewers’ comments

Reviewer 3

The review titled "Mitochondrial Integrity is critical in right heart failure development" delves into the molecular processes underlying right ventricular dysfunction (RVD) and right heart failure (RHF). Understanding these processes is crucial for developing tailored therapies that can mitigate the growing mortality rate among affected patients. This review highlights the emerging significance of mitochondrial dysfunction as a pivotal factor in RHF development. The objective of this comprehensive analysis is to summarize the current knowledge on mitochondrial dysregulation in both preclinical and clinical RVD and RHF.

The review is an outstanding scholarly work, characterized by meticulous attention to detail and well-documented analysis. It explores the topic comprehensively. The engaging writing style guides the reader through complex concepts. Well-written and documented!

In the introduction, it would be great if the authors could include a figure that visually illustrates the structure of mitochondria and highlights its multiple roles. This visual representation would effectively enhance the reader's understanding of the complex and diverse functions performed by mitochondria within cells. By providing a visual aid, the authors can succinctly convey the significance of mitochondria in various cellular processes, setting the stage for a comprehensive exploration of their role in the context of the review's focus.

In addition, it would greatly enhance the manuscript to include illustrative figures for key sections such as "Redox optimized ROS balance in the heart," "Mitochondrial integrity is impaired in right heart failure," and "Mitochondrial oxidative stress mediates right heart failure", “Therapeutic options to reduce mitochondrial oxidative stress in right heart failure”. The inclusion of visual aids would effectively complement the written analysis and facilitate comprehension of these critical aspects.

We very much appreciate the positive and constructive comments. We decided to combine the main aspects of our review in one comprehensive figure, which demonstrates the principal functions of the mitochondrial electron transport chain, OXPHOS and redox regulation, its interrelation with cardiomyocyte contractile function, and mitochondrial regulation and energy metabolism.

Round 2

Reviewer 1 Report

The authors have solved my concerns

Reviewer 3 Report

The authors have thoroughly addressed all of my concerns. No further comments.